# Extensive genomic study characterizing three *Paracoccaceae* populations and revealing *Pseudogemmobacter lacusdianii* sp. nov. and *Paracoccus broussonetiae* sp. nov.

Yang Deng,[1] Cong-Jian Li,[1] Jing Zhang,[2] Wei-Hong Liu,[3] Li-Yan Yu,[1] Yu-Qin Zhang[1]

**ABSTRACT** Bacteria within the family *Paracoccaceae* show promising potential for applications in various fields, garnering significant research attention. Three Gram stain-negative bacteria, strains CPCC 101601[T], CPCC 101403[T], and CPCC 100767, were isolated from diverse environments: freshwater, rhizosphere soil of *Broussonetia papyrifera*, and the phycosphere, respectively. Analysis of their 16S rRNA gene sequences, compared with those in the GenBank database, indicated that they belong to the family *Paracoccaceae*, with nucleotide similarities of 92.5%–99.9% to all of the *Paracoccaceae* members with valid taxonomic names. Phylogenetic studies based on 16S rRNA gene and whole-genome sequences identified CPCC 101601[T] as a member of the genus *Pseudogemmobacter*, CPCC 101403[T] belonging to the genus *Paracoccus*, and CPCC 100767 as part of the genus *Gemmobacter*. Notably, genomic analysis using average nucleotide identity (ANI; <95%) and digital DNA-DNA hybridization (dDDH; <70%) with their closely related strains suggested that CPCC 101601[T] and CPCC 101403[T] represent new species within their respective genera. Conversely, CPCC 100767 exhibited high ANI (98.5%) and dDDH (87.4%) values with *Gemmobacter fulvus* con5[T], indicating it belongs to this already recognized species. The in-depth genomic analysis revealed that strains CPCC 101601[T], CPCC 101403[T], and CPCC 100767 harbor key genes related to the pathways for denitrifying, MA utilization, and polyhydroxyalkanoate biosynthesis. Moreover, genotyping and phenotyping analysis confirmed that strain CPCC 100767 has the ability to convert atmospheric nitrogen into ammonia and produce 5-aminolevulinic acid, whereas CPCC 101601[T] can only perform the former bioprocess.

**IMPORTANCE** Based on polyphasic taxonomic study, two new species, *Pseudogemmobacter lacusdianii* and *Paracoccus broussonetiae,* affiliated with the family *Paracoccaceae* were identified. This expands our understanding of the family *Paracoccaceae* and provides new microbial materials for further studies. Modern genomic techniques such as average nucleotide identity and digital DNA-DNA hybridization were utilized to determine species affiliations. These methods offer more precise results than traditional classification mainly based on 16S rRNA gene analysis. Beyond classification of these strains, the research delved into their genomes and discovered key genes related to denitrification, MA utilization, and polyhydroxyalkanoate biosynthesis. The identification of these genes provides a molecular basis for understanding the environmental roles of these strains. Particularly, strain CPCC 100767 demonstrated the ability to convert atmospheric nitrogen into ammonia and produce 5-aminolevulinic acid. These bioprocess capabilities are of significant practical value, such as in agricultural production for use as biofertilizers or biostimulants.

**KEYWORDS** *Paracoccaceae*, genome, *Pseudogemmobacter lacusdianii*, *Paracoccus broussonetiae*, 5-aminolevulinic acid

Address correspondence to Yu-Qin Zhang, yzhang@imb.pumc.edu.cn.

Yang Deng and Cong-Jian Li contributed equally to this article. The author order was based on seniority.

The authors declare no conflict of interest.

See the funding table on p. 14.

The family *Rhodobacteraceae,* a member of the order Rhodobacterales*,* class Alphaproteobacteria*,* was initially identified by Garrity et al. (1) based on the phylogenetic analysis using the 16S rRNA gene sequences. The type genus of the family *Rhodobacteraceae* is *Rhodobacter*. The members of this family clustered into five phylogenetic lineages, namely *Rhodobacter*, *Paracoccus*, *Rhodovulum*, *Amaricoccus*, and *Roseobacter* (2). Recently, Hördt et al. (3) emended the description of the family *Rhodobacteraceae* according to the phylogenetic tree inferred from genome-scale data using the principles of phylogenetic systematics. Subsequently, the *Paracoccus* clade of the family *Rhodobacteraceae* was renamed as the family *Paracoccaceae*, with *Paracoccus* as the type genus (4). At the time of writing (May 2024), the family *Paracoccaceae* consisted of 68 validly described genera (https://lpsn.dsmz.de/family/paracoccaceae). Members of this family display considerable diversity in terms of phenotype, metabolism, and ecological adaptation and are predominantly distributed in marine (5, 6) and other aquatic ecosystems (7, 8) and in terrestrial saline-rich habitats (9–11).

The genus *Paracoccus*, the type genus and the largest lineage of the family *Paracoccaceae,* encompasses over 80 species across diverse ecosystems (https://lpsn.dsmz.de/genus/paracoccus). This group of bacteria is typically recognized as methylotrophs, utilizing C1 carbon compounds as substrates for polyhydroxyalkanoate (PHA) production, and demonstrates bioremediation potential characterized by its ability to degrade various environmental contaminants, like lindane, and deltamethrin (12–14).

In contrast, the genus *Pseudogemmobacter* represents a minor group in the family *Paracoccaceae* and contains only two validly described species (https://lpsn.dsmz.de/genus/pseudogemmobacter), while the genus *Gemmobacter* comprises 16 mainly phototrophic species (https://lpsn.dsmz.de/genus/gemmobacter).

In this study, strains CPCC 101601$^T$, CPCC 101403$^T$, and CPCC 100767 were isolated from freshwater samples, rhizosphere soil of *Broussonetia papyrifera*, and the phycosphere, respectively. Phenotypic tests and genome-based methods were used for their exact taxonomic characterization. All the results demonstrated that these strains affiliate with two undescribed and one described *Paracoccaceae* species. The comprehensive genomic studies indicated that CPCC 101601$^T$, CPCC 101403$^T$, and CPCC 100767 possess crucial genes associated with denitrifying, MA utilization, and PHA production pathways. In addition, extensive genotypic and phenotypic tests confirmed that strain CPCC 100767 possesses the capability for nitrogen (N) fixation and 5-aminolevulinic acid (5-ALA) biosynthesis, while CPCC 101601$^T$ carries the former process potential. It suggested that these novel strains could be useful in bioremediating environmental contamination such as treating wastewater.

## RESULTS

### Phenotypic features

CPCC 101601$^T$, CPCC 101403$^T$, and CPCC 100767 were all Gram srain-negative, rod-shaped bacteria. The cells of strain CPCC 101601$^T$ were motile with polar single flagellum, while those of strains CPCC 101403$^T$ and CPCC 100767 were nonmotile (Fig. S1). Strains CPCC 101601$^T$, CPCC 101403$^T$, and CPCC 100767 could grow on nutrient agar (NA), tryptic soy agar (TSA), and Reasoner's 2A agar (R2A) but demonstrated better growth on Peptone Yeast Glucose (PYG) and yeast extract-malt extract (YM). Colonies of strain CPCC 101601$^T$ on PYG agar were round, opaque, and raised, with glistening surfaces and complete edges, and were 1.0 mm–1.5 mm in diameter after 48 h of incubation at 30℃. Strain CPCC 101601$^T$ could grow in the presence of 0%–1% NaCl (optimum 0%) and at 20℃–37℃ (optimum 30℃). Strain CPCC 101601$^T$ grew over a pH range of 6.0–8.0, with an optimum pH of 7.0.

Colonies of strain CPCC 101403$^T$ formed on PYG agar were 0.8 mm–1.4 mm in diameter and were viscous, semitranslucent, circular, and convex after 48 h of cultivation at 30℃. CPCC 101403$^T$ grew at 10℃–37℃ (optimum, 30℃), pH 7.0–9.0 (optimum 7.0), and with 0%–5.0% (optimum, 0%–1.0%) NaCl.

Colonies of strain CPCC 100767 on PYG media incubated at 30°C for 48 h were 1.0 mm–1.7 mm in diameter, opaque, ivory, and convex with a smooth surface. Strain CPCC 100767 could grow at 4°C–37°C, pH 6.0–9.0, and in the presence of 0%–1% (wt/vol) NaCl; optimum growth was observed at 30°C, pH 7.0, and in the absence of NaCl, respectively.

The three strains exhibited catalase- and oxidase-positive reactivity but not starch hydrolysis, cellulose hydrolysis, gelatin hydrolysis, nitrate reduction, or $H_2S$ production; the Voges-Proskauer (VP) and methyl red (MR) test results were negative. All three strains demonstrated enzymatic activity for esterase lipase (C8) and naphthol-AS-BI-phospho-hydrolase but not lipase (C14), N-acetyl-β-glucosaminidase, α-chymotrypsin, α-fucosi-dase, α-galactosidase, α-glucosidase, α-mannosidase, β-galactosidase, β-glucosidase, and β-glucuronidase, according to API ZYM strip test. The major differentiating features among strains CPCC 101601$^T$, CPCC 101403$^T$, CPCC 100767, and other related species are shown in Table 1.

Strains CPCC 101601$^T$ and CPCC 100767 could grow in nitrogen-free Ashby medium, suggesting that these strains can fix atmospheric $N_2$. Furthermore, 5-ALA was detected in the fermentation broth of strain CPCC 101601$^T$. As demonstrated in Fig. S1, the linear regression equation $y = 0.0217x + 0.0161$, with an $R^2$ value of 0.9988, provided a satisfactory fit, and the 5-ALA content produced by strain CPCC 101601$^T$ in the fermentation broth was calculated to be 21.79 mg/L (Fig. S2).

## Chemotaxonomic properties

The common major fatty acid (>5% of the total fatty acids) in strains CPCC 101601$^T$, CPCC 101403$^T$, and CPCC 100767 was summed feature 8 ($C_{18:1}\omega 7c$ and/or $C_{18:1}\omega 6c$). $C_{16:0}$ was also detected in the major fatty acids of strains CPCC 101403$^T$ and CPCC 100767. Additionally, strain CPCC 100767 also contained $C_{18:1}\omega 7c$ 11-methyl and $C_{18:0}$ as the major fatty acids (Table 2). Strains CPCC 101601$^T$ and CPCC 101403$^T$ contained phosphatidylglycerol (PG), diphosphatidylglycerol (DPG), and phosphatidylcholine (PC) as the main components in the polar lipids profiles. While strain CPCC 101601$^T$ also contained phosphatidylethanolamine (PE) and phosphatidylmonomethylethanolamine (PME), strain CPCC 101403$^T$, instead, contained large amounts of unidentified glycolipids (GLs). The main polar lipid components of strain CPCC 100767 were PG, PE, and PC (Fig. S3). The primary respiratory quinone of strains CPCC 101601$^T$, CPCC 101403$^T$, and CPCC 100767 was Q-10.

## Primary identification of the newly isolated strains according to the 16S rRNA gene sequence

Nearly complete 16S rRNA gene sequences for strains CPCC 101601$^T$ (1,453 bp, accession number OR416956), CPCC 101403$^T$ (1,451 bp, accession number OR567423), and CPCC 100767 (1,455 bp, accession number OR417415) were obtained. Pairwise sequence comparison of the almost complete 16S rRNA gene in the EzBioCloud database revealed that strain CPCC 101601$^T$ is related to *Pseudogemmobacter hezensis* KCTC 82215$^T$ (97.4%), *Xinfangfangia humi* LMG 30636$^T$ (96.4%), and *Pseudogemmobacter bohemicus* DSM 103618$^T$ (96%). Strain CPCC 101403$^T$ is related to *Paracoccus yeei* ATCC BAA-599$^T$ (98.3%), *Paracoccus lutimaris* HDM-25$^T$ (98.3%), and *Paracoccus aestuariivivens* GHD-30$^T$ (97.7%). Strain CPCC 100767 is related to *Gemmobacter fulvus* JCM 34791$^T$ (99%) (Table S1). In the neighbor-joining phylogenetic trees constructed using the 16S rRNA gene sequences (Fig. 1), strains CPCC 101601$^T$, CPCC 101403$^T$, and CPCC 100767 formed a monophyletic clade with their related type strains, and these relationships were confirmed by the maximum-likelihood and maximum-parsimony methods (Fig. 1).

## Genome-based taxonomic relationships

The draft genome sizes of strains CPCC 101601$^T$, CPCC 101403$^T$, and CPCC 100767 were 3.8 Mb in 50 contigs, 4.8 Mb in 68 contigs, and 4.3 Mb in 32 contigs, respectively. The numbers of protein-coding, tRNA, rRNA, and other non-coding RNA (ncRNA)

**TABLE 1** Physiological characteristics of strains CPCC 101601[T], CPCC 101403[T], CPCC 100767, and closely related Paracoccaceae type strains[a]

| Characteristic | 1 | 2[b] | 3[c] | 4[d] | 5 | 6[e] | 7 | 8[f] |
|---|---|---|---|---|---|---|---|---|
| Isolation source | Dianchi Lake | Heavy metal-contaminated sludge | Bark samples of Populus × euramericana | Field soil | Rhizosphere soil of Broussonetia papyrifera | Human[e] | Phycosphere of Microcystis wesenbergii FACHB-908 | Culture system of Anabaena variabilis FBCC010004[f] |
| Temperature range (°C) | 4–37 | 20–32 | 15–37 | 4–45 | 10–37 | 25–42 | 20–37 | 4–37 |
| pH range | 6.0–8.0 | 7.0–8.0 | 6.0–10.0 | 5.5–9.0 | 7.0–9.0 | ND | 6.0–9.0 | 5.0–10.0 |
| NaCl tolerance (%, wt/vol) | 0–1 | 0–2 | 0–4 | 0–2 | 0–5 | 0–6 | 0–1 | 0–2 |
| Nitrate reduction | – | + | – | – | – | + | – | ND |
| API ZYM | | | | | | | | |
| Esterase (C4) | + | + | + | w | – | ND | + | + |
| Esterase lipase (C8) | + | + | + | w | + | ND | + | + |
| Lipase (C14) | + | + | w | – | – | ND | + | – |
| N-acetyl-β-glucosaminidase | – | – | + | – | + | ND | – | – |
| Trypsin | + | + | – | – | + | ND | – | – |
| α-Glucosidase | – | – | + | – | – | ND | – | + |
| β-Galactosidase | – | – | w | – | – | ND | – | – |
| Carbon sources used for growth | | | | | | | | |
| D-mannitol | + | + | + | – | – | + | + | + |

[a]Strains: 1, CPCC 101601[T]; 2, Pseudogemmobacter bohemicus Cd-10[T]; 3, Pseudogemmobacter hezensis D13-10-4-6[T]; 4, Xinfangfangia humi IMT-291[T]; 5, CPCC 101403[T]; 6, Paracoccus yeei ATCC BAA-599[T]; 7, CPCC 100767; 8, Gemmobacter fulvus con5[T]. +, positive; –, negative; w, weakly positive; ND, no data.
[b]Data were obtained from this study except Suman et al. (8).
[c]Data were obtained from this study except Ma et al. (15).
[d]Data were obtained from this study except Kämpfer et al. (16).
[e]Data were obtained from this study except Daneshvar et al. (17).
[f]Data were obtained from this study except Jin et al. (18).

**TABLE 2** Cellular fatty acid profiles of strains CPCC 101601[T], CPCC 101403[T], CPCC 100767, and their closely related strains[a]

| Strains | 1 | 2[b] | 3[c] | 4[d] | 5 | 6[e] | 7 | 8[f] | 9[g] | 10[h] |
|---|---|---|---|---|---|---|---|---|---|---|
| **Saturated** | | | | | | | | | | |
| $C_{16:0}$ | 3.6 | **19.9** | 5.4 | **10.5**[d] | **5.0** | **5.5** | **20.1** | **13.0** | **15.2** | **5.7** |
| $C_{17:0}$ | –[i] | – | – | – | – | – | – | 1.0 | – | – |
| $C_{18:0}$ | 4.1 | **26.3**[j] | 4.1 | **2.5**[d] | **5.1** | **7.6** | 1.6 | **5.0** | **6.8** | 1.6 |
| iso-$C_{15:0}$ | – | – | – | – | 1.5 | – | – | – | – | – |
| iso-$C_{16:0}$ | – | – | – | – | 1.9 | – | 1.0 | – | – | – |
| anteiso-$C_{15:0}$ | – | – | – | – | 1.7 | – | – | – | – | – |
| **Unsaturated** | | | | | | | – | | | |
| $C_{10:0}$ 3-OH | – | – | 2.4 | 2.8 | – | 3.0 | – | 3.0 | 2.5 | 2.8 |
| $C_{14:0}$ 3-OH | – | – | – | – | – | – | – | 1.0 | – | – |
| $C_{18:0}$ 3-OH | 4.6 | – | 2.6 | 3.3 | – | 2.1 | – | – | – | – |
| $C_{12:1}ω7c$ | – | – | – | – | – | – | – | 3.0 | – | – |
| $C_{16:1}ω7c$ | | 2.9 | 1.5 | – | – | – | – | – | – | – |
| $C_{17:1}ω8c$ | – | – | – | – | 1.3 | – | 1.0 | – | – | – |
| $C_{18:1}ω7c$ | – | 50.3 | 81.1 | 58.8 | – | – | – | 71.0 | – | 84.9 |
| $C_{18:1}ω9c$ | – | – | – | – | 2.4 | – | – | 1.0 | – | – |
| $C_{18:1}ω7c$ 11-methyl | – | – | – | 22.7 | 10.4 | 4.2 | – | – | – | – |
| $C_{19:0}$ cyclo $ω8c$ | – | – | – | – | – | – | 3.2 | – | – | – |
| **Summed features** | | | | | | | | | | |
| Sum In Feature 2 ($C_{14:0}$ 3-OH/iso-$C_{16:1}$ I) | – | – | – | – | – | – | 1.8 | – | 1.4 | 2.4 |
| Sum In Feature 3 ($C_{16:1}ω6c$/$C_{16:1}ω7c$) | 1.4 | – | – | – | 1.2 | 1.8 | 1.4 | – | – | 1.2 |
| Sum In Feature 4 (anteiso-$C_{17:1}$B/iso-$C_{17:1}$ I) | – | – | – | – | – | – | 3.0 | – | – | – |
| Sum In Feature 8 ($C_{18:1}ω6c$/$C_{18:1}ω7c$) | 84.3 | – | – | – | 62.4 | 74.5 | 63.3 | – | 73.4 | – |

[a]Strains: 1, CPCC 101601[T]; 2, *Pseudogemmobacter bohemicus* Cd-10[T]; 3, *Pseudogemmobacter hezensis* D13-10-4-6[T]; 4, *Xinfangfangia humi* IMT-291[T]; 5, CPCC 100767; 6, *Gemmobacter caeruleus* N8[T]; 7, CPCC 101403[T]; 8, *Paracoccus yeei* ATCC BAA-599[T]; 9, *Paracoccus lutimaris* HDM-25[T]; 10, *Paracoccus aestuariivivens* GHD-30[T].
[b]Data were from this study except Suman et al. (8).
[c]Data were from this study except Ma et al. (15).
[d]Data were from this study except Kämpfer et al. (16).
[e]Data were from this study except Qu et al. (19).
[f]Data were from this study except Daneshvar et al. (17).
[g]Data were from this study except Jung et al. (20).
[h]Data were from this study except Park et al. (21).
[i]–, Not detected.
[j]The bold values are the major components.

genes among the three novel strains were predicted and are listed in Table S2. Average nucleotide identity (ANI) and digital DNA-DNA hybridization (dDDH) values were calculated between strains CPCC 101601[T], CPCC 101403[T], and CPCC 100767 and their close relatives within the family *Paracoccaceae*. The ANI and dDDH values of strains CPCC 101601[T] and CPCC 101403[T] with their related strains were found to be lower than the currently recognized boundaries for genomic species definition (95–96% for ANI, 70% for dDDH) (Table S3). These results suggested that strain CPCC 101601[T] is a new species within the genus *Pseudogemmobacter*, and strain CPCC 101403[T] is a novel species within the genus *Paracoccus*. However, the overall genome relatedness indices between strains CPCC 100767 and *Gemmobacter fulvus* JCM 34791[T] were 98.5% for ANI and 78.4% for dDDH, showing that strains CPCC 100767 and JCM 34791[T] were the siblings of the species *Gemmobacter fulvus*, consistent with the results from the 16S rRNA gene analysis. The phylogenomic tree (Fig. S4) constructed according to the concatenated alignment of the core genes confirmed the results obtained from the phylogenetic analysis of the 16S rRNA gene sequence.

## Genes associated with key taxonomic characteristics

The metabolic processes of strains CPCC 101601[T], CPCC 101403[T], and CPCC 100767 use a specific system called type II fatty acid synthesis (FAS). This system is prevalent in most bacteria and plants, and each reaction of the FAS II pathway is catalyzed by a discrete enzyme. Acetyl-CoA carboxylase (ACC), which consists of four genes (*accA, accB, accC,*

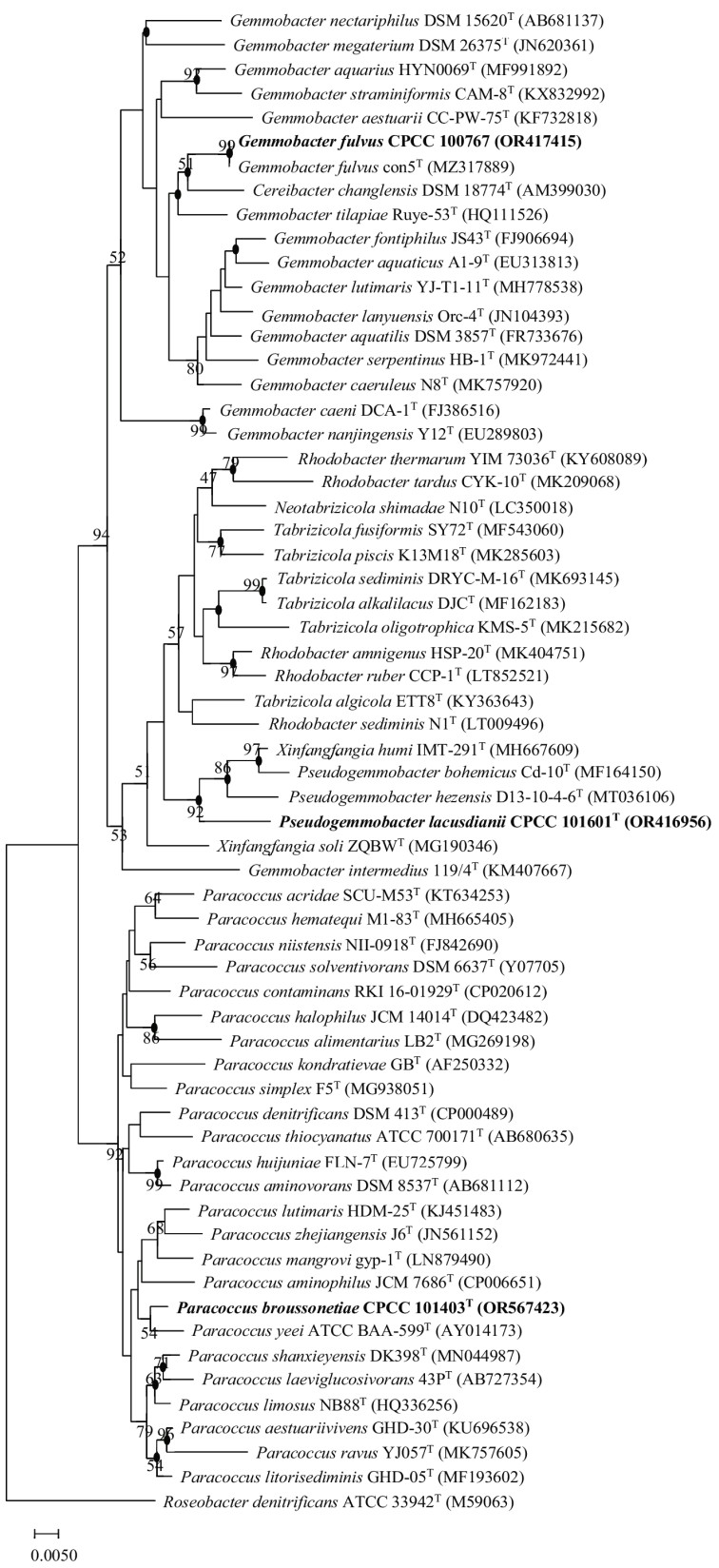

**FIG 1** Neighbor-joining tree constructed using 16S rRNA gene sequences showing the relationships between strains CPCC 101601^T, CPCC 101403^T, and CPCC 100767 and other representatives of the family *Paracoccaceae*. Filled circles indicate that the corresponding nodes were also recovered in

Fig 1 (Continued)

phylogenetic trees generated using the maximum-likelihood and maximum-parsimony methods. Bootstrap values above 50% are shown as percentages of 1,000 replicates. *Roseobacter denitrificans* ATCC 33942$^T$ (GenBank accession M59063) was used as the outgroup. The scale bar indicates 0.005 nt substitutions per alignment site.

and *accD*), uses biotin as a cofactor and bicarbonate as a substrate. In addition, ACC facilitates the ATP-dependent production of malonyl-CoA from acetyl-CoA (AcCoA) via $CO_2$ incorporation. Subsequently, malonyl-CoA is transferred to the acyl carrier protein (ACP) by malonyl-CoA/ACP transacylase (encoded by the *fabD* gene) (22). The enzymes of the fatty acid elongation cycle (encoded by *fabB, fabF, fabG, fabH, fabI, and fabZ*) are primarily responsible for the synthesis of long-chain acyl-[acyl carrier proteins], which are precursors of long-chain fatty acids (23). All the genomes of these strains contained genes (members of the Acc and Fab clusters) associated with fatty acid metabolism (FAS II system) (Table S4).

The major phospholipids of the three strains were constituted of PC, PG, DPG, PME, and PE. The biosynthesis of these phospholipids in prokaryotes starts with glycerol-3-phosphate which is sequentially converted to cytidine diphosphate diacylgly-cerol (CDP-DAG), the central precursor of all glycerophospholipids (24, 25). CDP-DAG is converted to PS (phosphatidylserine) by PS synthase (*pssA*) or PG by phosphatidylglgy-cerophosphate synthase (*pgpA*) and PG synthase (*pgsA*). In some specialized bacteria, CDP-DAG is used for PC biosynthesis via PC synthase (*pcs*) (26, 27). PG is converted to DPG (also denoted as cardiolipin) catalyzed by cardiolipin synthase (*cls*), and PS is decarboxylated to PE catalyzed by PS decarboxylase (*psd*). In some bacteria, PE serves as a precursor for PC synthesis by an S-adenosylmethionine-dependent methylation pathway [catalyzed by PE methyltransferase (*pemt*)] with the PME as the intermediates (28). The *pgsA*, *pgpA*, and *pcs*, which catalyze the steps in the synthesis of PG and PC, were retrieved from the genomes of strains CPCC 101601$^T$, CPCC 101403$^T$, and CPCC 100767 (Table S4). The genes associated with PE biosynthesis (*pssA* and *psd*) were present in the genomes of CPCC 101601$^T$ and CPCC 100767 but not in CPCC 101403$^T$, although the *pemt* (response to catalyze PE to PME and PC) was found in all genomes. Moreover, the gene *cls*, which catalyzed DPG synthesis via transphosphatidylation of two PG molecules, was only found in CPCC 101403$^T$.

The benzenoid aromatic nucleus of quinone is derived from the shikimate pathway. The ubiquinone (abbreviation, Q) biosynthetic pathway branches from the shikimate pathway at the chorismite level, and the biosynthesis of Q depends on the *ubi* gene. UbiA catalyzes the prenylation of 4-hydroxybenzoate, and three of the reactions in the biosynthesis of Q involve hydroxylation, leading to the introduction of hydroxyl groups at positions C-6, C-4, and C-5 of the benzene nucleus of Q. The hydroxylation reactions are catalyzed by the products of *ubiB*, *ubiH*, and *ubiF*. In addition, methylation and hydroxylation reactions occur alternately and are catalyzed by methyltransferases encoded by *ubiG* and hydroxylases encoded by *ubiE* (29). *UbiX* is thought to be involved in the decarboxylation step in Q biosynthesis (30). *UbiA*, *ubiE*, and *ubiH* genes were present in the genomes of strains CPCC 101601$^T$, CPCC 101403$^T$, and CPCC 100767, while the *ubiX* gene was identified in the genomes of strains CPCC 101601$^T$ and CPCC 101403$^T$, and the *ubiB* gene was retrieved from the genome of strain CPCC 101403$^T$ (Table S4).

## Key features involved in wastewater treatment according to the genomic analysis

Denitrifying phosphorus-accumulating organisms (DPAOs) have been applied to remove inorganic nitrogen and phosphorus from sewage and other industrial waste streams for wastewater treatment (31). The mechanism of phosphorus (P) and N removal by these strains involves two metabolic pathways: denitrification and poly-P biosynthesis. For denitrification, DPAOs can use dissolved exogenous N (nitrate and nitrite) as an electron acceptor and subsequently return N to the atmosphere in the form of nitrogen (32). Four

kinds of enzymes, nitrate reductase (*nar*), nitrite reductase (*nir*), nitric oxide reductase (*nor*), and nitrous oxide reductase (*nos*), are involved in this catalytic process (33). Cells absorb phosphate and convert it into poly-P for storage using polyphosphate kinase (*PPP*). In addition, some DPAOs can transport volatile fatty acids into cells and activate them to produce AcCoA via AcCoA synthase. Subsequently, AcCoA is transformed into PHA by three enzymes: β-ketothiolase (*phaA*), acetoacetyl-CoA reductase (*phaB*), and polymerase (*phac*) (12). Several strains from the family *Paracoccaceae* have been reported to accomplish simultaneous N and P removal (33). Accordingly, we identified genes related to denitrifying P removal in the genomes of the identified novel strains (Table S5). Interestingly, strain CPCC 100767 contained the most genes related to denitrification, while strain CPCC 101601$^T$ did not demonstrate any of these genes. Strain CPCC 101403$^T$ was found to possess genes involved in the dissimilatory effect of exogenous nitrate instead of those involved in denitrification genes. Moreover, strains CPCC 101403$^T$ and CPCC 100767, which harbor *phaA* and *phaC*, could biosynthesize PHA.

In addition to eutrophication, contamination by methylamine has garnered significant interest due to its potent toxicity to aquatic organisms and the broader environment (34). Methylamine removal by microorganisms has emerged as a prevalent approach to wastewater treatment. Interestingly, multiple *Gemmobacter* strains capable of degrading methylamine, such as *Gemmobacter* sp. LW-1, *Gemmobacter lutimaris*, and *Gemmobacter caeni,* have been reported (35). The methylamines in wastewater include trimethylamine (TMA), dimethylamine (DMA), and monomethylamine (MMA). Cells transform TMA into DMA using the enzymes TMA dehydrogenase (*tdm*), TMA monooxygenase (*tmm*), or TMA methyltransferase (*mttB*) (36). Subsequently, under the action of dehydrogenase (dmd) or DMA monooxygenase (*dmmDABC*), DMA is degraded to MMA (37). Two alternative pathways are involved in MMA oxidation: the N-methylglutamate (NMG) pathway and the direct MMA oxidation pathway. Several enzymes involved in these two pathways are listed in Table S5. Here, genes associated with methylamine metabolism were identified using Rapid Annotation using Subsystem Technology (RAST) (Table S5). Notably, most methylamine metabolism-related genes were present in the genomes of the three studied strains. Strains CPCC 101601$^T$ and CPCC 100767 do not possess the gene related to MMA oxidation. Strain CPCC 100767 lacks the *mttB* gene, which is responsible for TMA degradation, but expresses an alternative gene, *tmm*. The distribution of genes related to methylamine metabolism suggested that the three novel strains may be able to degrade methylamine on a genetic basis, indicating their potential application in methylamine removal for wastewater treatment.

By supplementing the nitrogen requirements of suspended growth systems, bacterial nitrogen fixation can significantly improve the efficiency of treating wastewater with a high carbon-to-nitrogen ratio (such as wastewater from paper mills) (38). $N_2$-fixing bacteria are characterized by the presence of the *nifH* gene, which encodes the iron protein subunit of the nitrogenase enzyme complex responsible for biological nitrogen fixation (39). As previously mentioned, strains CPCC 101601$^T$ and CPCC 100767 could fix atmospheric $N_2$ in nitrogen-fixing experiments, while CPCC 101403$^T$ could not. Consistently, the *nifH* gene was present in the genomes of strains CPCC 101601$^T$ and CPCC 100767 but not in that of CPCC 101403$^T$ (Table S5). These findings suggested that strains CPCC 101601$^T$ and CPCC 100767 possess genotypes and phenotypes associated with nitrogen fixation.

Several studies have demonstrated that various valuable compounds, including ALA, such as 5-ALA, can be released from photosynthetic bacteria cells during wastewater treatment (40). Due to its biochemical characteristics, 5-ALA, an important intermediate in tetrapyrrole biosynthesis in organisms, has been widely applied in many fields, such as medicine, agriculture, and the food industry. Two natural pathways for ALA biosynthesis have been reported: C4 and C5. Due to its relatively lower complexity than the C5 pathway, the C4 pathway exhibits a greater capacity for attaining elevated ALA synthesis. The C4 pathway, also known as the Shemin pathway, is responsible for the synthesis of 5-ALA using succinyl-CoA (a C4-compound) and glycine through a single-step

decarboxylating condensation reaction involving ALA synthases (ALAS, *hemA*) (41). In this study, the fermentation analysis revealed that only strain CPCC 101601[T] could produce ALA. Consistently, the *hemA* protein encoding ALA synthase was detected only in the genome of strain CPCC 101601[T] (Table S5).

Overall, the strains obtained in this study can potentially remove N and P simultaneously. Additionally, Strain CPCC 101601[T] can simultaneously synthesize import byproducts, such as ALA. Consequently, the three newly isolated strains can potentially be beneficial in the future development of a new bioremediation process.

## Secondary metabolite biosynthesis gene clusters analysis

The findings collected from the antibiotics and Secondary Metabolite Analysis Shell (antiSMASH) indicated that strains CPCC 100767, CPCC 101403[T], and CPCC 101601[T] possess the 8, 11, and 3 clusters of secondary metabolite genes, respectively (Table S6). These gene clusters exhibited 6%–100% similarities to previously documented secondary metabolite biosynthetic gene clusters, such as cupriachelin, JBIR-06, parabactin, oryzanaphthopyran A/oryzanaphthopyran B/oryzanaphthopyran C/ oryzanthrone A/oryzanthrone B/chlororyzanthrone A/chlororyzanthrone B gene clusters, and other unidentified secondary metabolite clusters attributable to nonribosomal peptide synthetase (NRPS), NRPS-like, thioamide-NRP, redox-cofactor, Type I polyketide synthases (T1PKS), Type III polyketide synthases (T3PKS), post-translationally modified peptides (RiPP)-like, hserlactone, terpene, β-lactone, thioamides types, and hydrogen-cyanide, respectively (Table S6).

## DISCUSSION

Plateau lakes are essential aquatic ecosystems that play an essential role in the biogeochemical cycles, especially carbon, nitrogen, and phosphorus cycles of continental watersheds (42). Microorganisms thriving in the lakes drive these cycles, which is crucial to maintaining proper ecosystem function. In contrast, the microbial community structure is also shaped by the nutrient availability in the lake ecosystems (43). Dianchi Lake, located on the Yunnan Plateau, is the sixth largest freshwater lake (44). The water quality of Dianchi Lake is constantly deteriorating as a result of the pollution loading (45). Moreover, sufficient sunlight and eutrophication usually indicate a high risk of photosynthetic bacterioplankton (like cyanobacteria and alga) bloom in Dianchi Lake. Therefore, aerobic nitrogen-fixing is the main process for fixing atmospheric $N_2$ and participates in nitrogen cycling in the oxygen enrichment environment (46). Here, a novel *Pseudogemmobacter* strain CPCC 101601[T] was isolated from a freshwater sample from Dianchi Lake. Phenotypic and genotypic analyses show that it has the capability for MA degradation and nitrogen fixation. Unexpectedly, the full set of genes involved in the denitrification (use dissolved exogenous nitrate and/or nitrite as an electron acceptor and subsequently return N to the atmosphere in the form of $N_2$) was absent in the genome of CPCC 101601[T]. It suggested that strains CPCC 101601[T] participate in organic nitrogen cycling by MA degradation in the eutrophication lake ecosystem. In contrast, strain CPCC 100767, isolated from the culture system of a cyanobacterial strain *Microcystis wesenbergii* FACHB-908, exhibits the characteristics related to the entire process of nitrogen cycling (encompassing the organic and inorganic nitrogen), including nitrogen fixation, MA utilization, and denitrification. Given that the cyanobacterial strain was obtained from the Freshwater Algae Culture Collection, strain CPCC 100767 might originally colonize a freshwater ecosystem and play an essential role in the nitrogen cycle.

The rhizosphere is an intricate environment that harbors a strikingly diverse microbial community (47). Ecological theories propose that the plant-associated microbial community is shaped by complex interactions among the host, microorganisms, and the environment. The plant-associated microbes can provide benefits to the plant through various direct or indirect mechanisms. Direct effects are mediated through fixing atmospheric nitrogen, unlocking of essential nutrients from minerals,

and enhancing the capability of plants to absorb nutrients from the soil. Effects can be indirect, as the plant-associated microbes protect the plant against pathogens or pests through antagonism or through inducing systemic resistance in plants (48). The novel rhizospheric strain CPCC 101403$^T$ lacks the genes responsible for these pathways in terms of both phenotype and genotype but possesses the genes associated with the biosynthesis of PHA and dissimilation of nitrogen. Additionally, various unidentified biosynthetic gene clusters encoding hserlactone, NRPS-like, and polyketide synthase (PKSs) were present in the genome of CPCC 101403$^T$. These gene clusters may be responsible for disease-suppressive functions in the plant-associated microbes (49, 50). The aforementioned observations demonstrated that strain CPCC 101403$^T$ might benefit the host plant indirectly by producing the antibiotic against pathogens.

## Description of the new taxon and diagnostic traits

### *Pseudogemmobacter lacusdianii sp. nov.*

*Pseudogemmobacter lacusdianii* sp. nov. (*la.cus.di.a'nii*. L. n. lacus, -us, lake; Dian, name of a lake; N.L. gen. n. *lacusdianii* from Dian Lake from where the bacterium was isolated).

The cells are Gram stain-negative, aerobic, rod-shaped, and motile. Colonies on PYG agar media are round, opaque, and raised, with glistening surfaces and complete edges, and are 1.0 mm–1.5 mm in diameter after 48 h of incubation at 28℃. The optimal growth temperature is 28℃, and the optimum pH range is 7.0. The bacterium grows in the presence of 1% (wt/vol) NaCl (optimum 0%). The bacterium does not reduce nitrate or produce H$_2$S; the VP and MR test results were negative. *Pseudogemmobacter lacusdianii* sp. nov. can produce 5-ALA. This strain is positive for oxidase, catalase, esterase (C4), lipase esterase (C8), leucine arylamidase, naphthol-AS-BI-phosphohydrolase, trypsin, and valine arylamidase. The strain does not hydrolyze skim milk, gelatin, or starch. The strain can assimilate the following carbohydrates: citric acid, D-arabitol, D-fructose, D-manni-tol, D-mannose, D-saccharic acid, formic acid, glucuronamide, glycerol, glycyl-L-proline, inosine, L-alanine, L-arginine, L-glutamic acid, L-histidine, L-lactic acid, L-malic acid, L-serine, methyl pyruvate, Tween 40, α-D-glucose, *p*-hydroxyphenylacetic acid, and γ-amino-butyric acid. The prevailing respiratory quinone is Q-10, the primary cellular fatty acid is summed feature 8 (C$_{18:1}$ ω7c and/or C$_{18:1}$ ω6c), and the main polar lipid components are phatidylglycerol (PG), diphosphatidylglycerol (DPG), phosphatidylcho-line (PC), phosphatidylethanolamine (PE) and phosphatidylmonomethylethanolamine (PME). The type strain CPCC 101601$^T$ (= AB283$^T$ =KCTC 8066$^T$) was isolated from a water sample from Dianchi Lake, China. The G + C content of the genomic DNA of the type strain is 61.0%. The 16S rRNA gene and whole-genome sequence of strain CPCC 101601$^T$ are publicly available through the accession numbers OR416956 and JAVDBT000000000, respectively.

### *Paracoccus broussonetiae sp. nov.*

*Paracoccus broussonetiae* sp. nov. (brous.so.ne'ti.ae. N.L. gen. n. *broussonetiae*, of *Broussonetia*, the name of the plant genus, referring to the isolation of the type strain from the rhizosphere of a plant *Broussonetia papyrifera*).

The cells are Gram stain-negative, aerobic, rod-shaped, and nonmotile. Colonies on PYG agar media are 0.8 mm–2.0 mm in diameter and are viscous, semitranslucent, circular, and convex after 48 h of cultivation at 30℃. Growth occurs at 10℃–37℃ (optimum, 30℃), pH 7.0–9.0 (optimum 7.0), and with 0%–5.0% (optimum, 0%–1.0%) NaCl. The bacterium demonstrates catalase and oxidase activities and does not hydrolyze gelatin, starch, or cellulose. The bacterium does not reduce nitrate or produce H$_2$S; the VP and MR test results were negative. Growth is supported by 3-methyl glucose, acetic acid, D-arabitol, D-fructose, D-fructose-6-PO$_4$, D-fucose, D-galactose, D-galactur-onic acid, D-gluconic acid, D-glucuronic acid, D-malic acid, D-mannose, formic acid, glucuronamide, glycyl-L-proline, inosine, L-alanine, L-aspartic acid, L-fucose, L-galac-tonic acid lactone, L-glutamic acid, L-histidine, L-lactic acid, L-malic acid, L-rhamnose,

L-serine, methyl pyruvate, mucic acid, myo-Inositol, N-acetyl-D-galactosamine, *N*-acetyl-D-glucosamine, quinic acid, *α*-D-glucose, *α*-keto-glutaric acid, *β*-hydroxy-D,L-butyric acid, and *γ*-amino-butyric acid. Substrates that do not support growth include bromo-succinic acid, citric acid, dextrin, D-cellobiose, D-glucose-6-$PO_4$, D-lactic acid methyl ester, D-maltose, D-mannitol, D-melibiose, D-raffinose, D-salicin, D-serine, D-sorbitol, D-trehalose, D-turanose, gelatin, gentiobiose, L-arginine, L-pyroglutamic acid, N-acetyl neuraminic acid, pectin, *p*-hydroxyphenylacetic acid, propionic acid, stachyose, sucrose, Tween 40, α-D-lactose, α-hydroxybutyric acid, α-ketobutyric acid, and β-methyl-D-gluco-side. Q-10 is the only isoprenoid quinone in this strain. DPG, PG, PC and unidentified GLs are the strain's main polar lipids. The major fatty acids are summed feature 8 ($C_{18:1}ω7c$ and/or $C_{18:1}ω6c$) and $C_{16:0}$. The type strain CPCC 101403$^T$ (= B328$^T$ =KCTC 8291$^T$) was isolated from rhizosphere soil of *B. papyrifera* in Yunnan Province, China. The genomic DNA G + C content of the type strain is 64.0%. The 16S rRNA gene and whole-genome sequence of strain CPCC 101403$^T$ are publicly available through the accession numbers OR567423 and JAVRQI000000000, respectively.

## Conclusion

Three newly isolated *Paracoccaceae* strains affiliated with three different genera were characterized. Polyphasic taxonomic studies revealed two novel species, *Pseudogem-mobacter lacusdianii* sp. nov., with CPCC 101601$^T$ as the type strain, and *Paracoccus broussonetiae* sp. nov., with CPCC 101403$^T$ as the type strain. The phycosphere-derived strain CPCC 100767 was identified as a non-type strain of *Gemmobacter fulvus*. The genes related to the pathways in wastewater treatment, containing denitrifying, MA utilization, and PHA biosynthesis, were variously distributed in the genome of these novel strains. Furthermore, strains CPCC 101601$^T$ and CPCC 100767 have the capability to fix the atmospheric $N_2$ based on the genotype and phenotype. Noticeably, the fermentation experiments demonstrated that only CPCC 101601$^T$ can produce the high-level 5-ALA (21.79 mg $L^{-1}$), a precursor of molecules such as hemoglobin and vitamin $B_{12}$. Consistently, the *hemA* gene, encoding 5-aminolevulinic acid synthetase, which catalyzes the formation of 5-ALA, was only found in the genome of strain CPCC 101601$^T$. In conclusion, these novel strains have potential applications for the bioremediation of wastewater, especially eutrophication and methylamine contamination.

## MATERIALS AND METHODS

### Sampling and bacteria isolation

Strain CPCC 101601$^T$ was obtained from a water sample procured from Dianchi Lake (25°26'51'' N, 102°42'48'' E), the greatest plateau freshwater lake in Kunming, Yunnan Province, southwestern China. The water sample (500 mL) was filtered, and bacterial strains were isolated according to the procedure described by Gong et al. (51).

Strain CPCC 101403$^T$ was recovered from the rhizosphere soils of *B. papyrifera* growing in Chuxiong (25°02'30.06''N, 101°54'02.38''E), central Yunnan Province. Rhizosphere soil was gently removed from the roots using a sterile steel shovel. Two grams of wet soil were mixed with 18 mL of 0.85% (wt/vol) NaCl solution to produce a $10^{-1}$ diluted soil suspension. Subsequently, 2 mL of the $10^{-1}$ diluted soil suspension was transferred to 18 mL of newly prepared 0.85% (wt/vol) to produce a $10^{-2}$ diluted soil suspension. Then, 200 µL of $10^{-4}$ diluted soil suspensions was spread on PYG agar plates (g/L; peptone 3, yeast extract 5, glycerol 10, betaine hydrochloride 1.25, sodium pyruvate 1.25, and agar 15, pH 7.2). The isolation plates were incubated aerobically at 30°C, allowing for the isolation of distinct colonies. Distinct uniform colonies were picked from the isolation plates and streaked onto newly prepared PYG agar plates.

Strain CPCC 100767 was recovered from a culture system of *Microcystis wesenbergii* FACHB-908 in our laboratory. Initially, the cyanobacterial strain *Microcystis wesenbergii* FACHB-908 was obtained from the Freshwater Algae Culture Collection at the Institute of

Hydrobiology (FACHB collection). Bacterial strains were acquired as described by Zhang et al. (52).

Purified strains CPCC 101601$^T$, CPCC 101403$^T$, and CPCC 100767 were cultivated on PYG slants at 30°C. For long-term storage, glycerol suspensions (20%, vol/vol) of the strain were prepared and stored at –80°C.

## Growth condition test and phenotypic characterization

Bacterial growth conditions were tested on YM (Difco), TSA (Difco), R2A (Difco), NA, and PYG agar media. The tested strains were incubated at different temperatures (4, 10, 15, 20, 25, 28, 30, 32, 35, 37, 40, and 45°C), pH values (4.0–11.0 in 1.0 intervals), and NaCl concentrations [0%–10% (w/v)] in sodium-free PYG broth to assess their tolerances to these factors. The morphological properties of the colonies were recorded after 48 h of incubation on PYG agar at 30°C. The Gram reaction was performed as described by Magee et al. (53). Cell motility was examined by light microscopy using the hanging drop method (54) (Axio A1 Vario, Zeiss). A catalase test was conducted using 3% (vol/vol) hydrogen peroxide, and oxidase activity was determined using the API oxidase reagent (bioMérieux). The cellulose degradation activity was examined using carboxy-methylcellulose-sodium screening medium. The ability to produce $H_2S$, reduce nitrate, and hydrolyze gelatin and starch were assessed with the corresponding kits (HuanKai Microbial) according to the manufacturer's instructions. VP and MR tests were conducted using glucose-peptone broth media. The ability of the tested strains to assimilate carbon compounds was examined at 30°C using Biolog GEN III Microplates. The results were observed and analyzed using an Omnilog device (Biolog Inc., Hayward, CA, USA). Other biochemical and physiological parameters were determined using API ZYM and API 50CH test kits (bioMérieux) according to the manufacturer's instructions. The results were evaluated after incubation at 30°C for 3 days.

The nitrogen fixation capacity was verified using Ashby media. Additionally, the 5-ALA biosynthesis capacity of strains CPCC 101601$^T$, CPCC 101403$^T$, and CPCC 100767 was assessed as follows: strains CPCC 101601$^T$, CPCC 101403$^T$, and CPCC 100767 were inoculated into Molisch liquid media (g/L; peptone 10, glycerin/dextrin 5, $MgSO_4$ 0.5, $KH_2PO_4$ 0.5, $FeSO_4$ trace, pH 7.2) and cultured at 180 rpm at 30°C for 7 days. The culture solution was centrifuged at 4,000 rpm for 10 min, and the supernatant was collected. A 500 µL sample or standard solution was mixed well with 500 µL of 1 M acetate buffer (pH 4.6) and 100 µL of acetylacetone. Subsequently, the resulting mixture was heated at 100°C for 10 min. After cooling to room temperature, a 100 µL portion of the reaction mixture was combined with 100 µL of freshly prepared, modified Ehrlich's reagent in a 96-well plate, ensuring minimal exposure to light. Finally, the absorbance at 553 nm was measured using a spectrophotometer (SpectraMax 340, Molecular Devices) after a 10-minute reaction (55).

## Chemotaxonomic characteristics

The cells were collected when the cultures reached approximately 70% of their maximal optical density during the exponential growth phase. To analyze cellular fatty acids, 40 mg of bacterial cells were exposed to a series of four different reagents. This exposure was followed by saponification and methylation of the fatty acids to detach them from lipids. The resulting fatty acid methyl esters were then analyzed using gas chromatography. The peaks were labeled, and the corresponding chain length values were determined using established methods (MIDI, version 6.0). The TSBA 6 database was used for the analysis (56, 57). Cellular polar lipids were extracted, detected using two-dimensional thin layer chromatography (TLC), and identified (58). The respiratory isoprenoid quinone was prepared using a chloroform/methanol mixture (2:1, vol/vol). The extract was then evaporated under a vacuum and further extracted with acetone. The extracted quinone was subsequently analyzed using high-performance chromatography (59).

## 16S rRNA gene sequencing and phylogenetic analysis

Genomic DNA was extracted from strains CPCC 101601[T], CPCC 101403[T], and CPCC 100767, and the 16S rRNA genes were amplified using the universal bacterial primers 27F (5′-AGAGTTTGATCCTGGCTCAG-3′) and 1492R (5′- GGTTACCTTGTTACGACTT-3′) (60). Gene similarities to previously documented type strains were ascertained using the EzBio-Cloud server (61) (https://www.ezbiocloud.net). The phylogenetic trees were constructed using MEGA 11 software (62) according to the neighbor-joining method (63), maximum-parsimony algorithm (64), and maximum-likelihood (65) algorithm. The evolutionary distances were calculated using the Kimura 2-parameter model (66), and the statistical reliability of these trees was assessed using bootstrap analysis with 1,000 replications (67).

## Whole-genome sequencing, assembly, and annotation

Whole-genome sequencing was conducted using an Illumina HiSeq 4,000 system (Illumina, San Diego, CA, USA) at the Beijing Genomics Institute (Shenzhen, China). Illumina PCR adapter reads and low-quality reads from the paired-end library were filtered. All good-quality paired reads were assembled using SPAdes (68) into several scaffolds. Scaffolds shorter than 500 bp were excluded. The genome completeness and contamination assessments were performed using the CheckM pipeline (69). The ANI and dDDH values between the genomes of three novel strains and their closely related type strains were calculated using the OrthoANI tool (https://www.ezbiocloud.net/tools/orthoani) and the Genomic-to-Genomic Distance Calculator (3.0; http://ggdc.dsmz.de/ggdc.php) (recommended formula-2.0) (70), respectively. The G + C content of the DNA was calculated using whole-genome sequences. Additionally, a concatenate-based phylogenomic analysis was performed on CPCC 101601[T], CPCC 101403[T], CPCC 100767, and their related *Paracoccaceae* family type strains using the EasyCGTree pipeline (https://github.com/zdf1987/EasyCGTree4) (71). IQ-TREE (v.1.6.1) (72) was used for phylogenetic tree inference.

The assembled genomes of strains CPCC 101601[T], CPCC 101403[T], and CPCC 100767 were subjected to protein-coding sequence (CDS) prediction using the NCBI Prokaryotic Genome Annotation Pipeline. The putative CDS were subjected to functional annotation using the Rapid Annotation server (https://rast.nmpdr.org). Gene cluster prediction for natural products was conducted using antiSMASH (http://antismash.secondarymetabolites.org).

## ACKNOWLEDGMENTS

This research was supported by National Natural Science Foundation of China (32170021), Beijing Natural Science Foundation (5212018), CAMS Innovation Fund for Medical Sciences (CIFMS, 2021-I2M-1-055), Key Project at Central Government Level - the ability establishment of sustainable use for valuable Chinese medicine resources (2060302), and the National Infrastructure of Microbial Resources (NIMR-2023-3).

Y.D., C.-J.L., and J.Z. carried out the experiments and data analysis; C.-J.L., J.Z., and L.-Y.Y. collected the environmental samples; Y.D., and Y.-Q.Z. conceived the research, analyzed the data, and prepared the manuscript. All authors contributed to the article and approved the submitted version.

## AUTHOR AFFILIATIONS

[1]Institute of Medicinal Biotechnology, Chinese Academy of Medical Sciences & Peking Union Medical College, Beijing, China
[2]Southern Marine Science and Engineering Guangdong Laboratory, Guangzhou, China
[3]Yunnan Provincial Key Laboratory of Entomological Biopharmaceutical R&D, Dali University, Dali, China

## AUTHOR ORCIDs

Yu-Qin Zhang http://orcid.org/0000-0002-1739-6705

## FUNDING

| Funder | Grant(s) | Author(s) |
|---|---|---|
| MOST | National Natural Science Foundation of China (NSFC) | 32170021 | Yu-Qin Zhang |

## AUTHOR CONTRIBUTIONS

Yang Deng, Data curation, Investigation, Methodology, Resources, Validation, Writing – original draft | Cong-Jian Li, Investigation, Resources, Validation, Writing – original draft | Wei-Hong Liu, Data curation, Formal analysis, Investigation, Resources, Validation | Li-Yan Yu, Investigation, Validation | Yu-Qin Zhang, Conceptualization, Formal analysis, Funding acquisition, Project administration, Supervision, Validation, Writing – review and editing.

## ETHICS APPROVAL

This research did not contain any studies with animals performed by any of the authors.

## ADDITIONAL FILES

The following material is available online.

### Supplemental Material

**Supplemental material (Spectrum01088-24-s0001.docx).** Tables S1 to S4; Fig. S1 to S6.

### Open Peer Review

**PEER REVIEW HISTORY (review-history.pdf).** An accounting of the reviewer comments and feedback.

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
