## [Reviewer comments · Microbiology Spectrum]

Microbiology Spectrum

Extensive genomic study characterizing three *Paracoccaceae* populations and revealing *Pseudogemmobacter lacusdianii* sp. nov. and *Paracoccus broussonetiae* sp. nov.

Ynag Deng, Congjian Li, Jing Zhang, Wei-Hong Liu, Li-Yan Yu, and Yu-Qin Zhang

Corresponding Author(s): Yu-Qin Zhang, Chinese Academy of Medical Sciences & Peking Union Medical College

Review Timeline:

Submission Date:	April 30, 2024
Editorial Decision:	June 3, 2024
Revision Received:	June 27, 2024
Editorial Decision:	July 4, 2024
Revision Received:	July 10, 2024
Accepted:	July 18, 2024

Editor: Eva Sonnenschein

Reviewer(s): Disclosure of reviewer identity is with reference to reviewer comments included in decision letter(s). The following individuals involved in review of your submission have agreed to reveal their identity: Stephen A Jackson (Reviewer #2)

Transaction Report:

DOI: <https://doi.org/10.1128/spectrum.01088-24>

Re: Spectrum01088-24 (**Extensive genomic study characterizing three *Paracoccaceae* populations and revealing *Pseudogemmobacter lacusdiani* sp. nov. and *Paracoccus broussonetiae* sp. nov.**)

Dear Dr. Yu-Qin Zhang:

Thank you for the privilege of reviewing your work. Below you will find my comments, instructions from the Spectrum editorial office, and the reviewer comments.

Revision Guidelines

Sincerely,
Eva Sonnenschein
Editor
Microbiology Spectrum

Reviewer #1 (Comments for the Author):

The manuscript submitted by Deng Yang, et al. described two new species in family of Paracoccaceae, based on phylogenetic, chemotaxonomic and phenotypic analyses. The polyphasic analysis supports the assignment of these strains to be the novel species. Analysis genes associated with key taxonomic characteristics is a good model to merge strain's phenotype and its genome, which will be followed by the other taxonomists.

And also the authors characterizes their abilities of denitrification, MA utilization and PHA biosynthesis. This will supply a deeply understanding on their ecologic roles and potential utilization. I think it is fit to this journal to publishing. While some questions should be discussed or revised before accepted.

Firstly, analysis genes associated with key taxonomic characteristics is very important, authors have clarified the genes related to the ubiquinone, fatty acids and phospholipid PC, PG, while the other lipids also important, especial the diagnostic lipids such as PE, DPG and PME. Whether authors can supply the data of biosynthesis genes related these lipids in part of "Genes associated with key taxonomic characteristics".

Secondary, based genomic and physiologic data, three strains' potential ecological function in original isolation environment should be discussed.

Some minor revisions should be revised as follow.

1. Line 105, delete "but not in the presence of 2% NaCl" and "but not 40 °C".
2. Data in table S1 is important, so table S1 should be moved to the text, and the same component in Sum in feature lines and the other lines also should be revised or merged.
3. Many Abbreviation names were not listed in the Abbreviations part in line 56, please supply them.
4. 5-aminolevulinic acid is not a product related wastewater treatment, so suggest move in lines 248-259 to the part of "Secondary metabolite biosynthesis gene clusters analysis".
5. Deposition certificates of two type strains should be supplied as supplement files.

Reviewer #2 (Comments for the Author):

General Comments

The authors isolated 3 bacterial strains, from freshwater, from the rhizosphere soil of *Broussonetia papyrifera*, and from the phycosphere of *Microcystis wesenbergii*. The isolates of the Family Paracoccaceae were characterized genomically and phenotypically. Two of the isolates were designated as the type strains of novel species.

The manuscript is well written and well structured. The methods employed were thorough and appropriate. The focus is interesting and of interest to the general reader.

Have specimens of the newly described species been deposited to two Culture Collections in different countries?

A small number of minor corrections are recommended.

Specific Comments

Line 62: Taxonomic names at the level of Family and below should be italicized, ranks above Family should not be italicized.

Line 70: at the time of review this number is 68.

Line 87: '*Broussonetia*'.

Line 171: 'species'.

Lines 189-90: The phrase 'showing few inconspicuous disparities' is vague and the meaning is unclear.

Line 232: '*Gemmobacter*'.

Line 302: The proposed new species name is '*lacusdiani*': this will need input from a Latin expert but I believe, based on my understanding of the nomenclature rules, that this should be '*lacusdianii*'.

Lines 315 & 339 'pyruvate'.

Line 427 & Figure 1: It is curious and unusual that in the Microsoft Word version of the manuscript. all elements of Figure 1 (species names on each 'leaf', bootstrap values etc.) appear as individual ungrouped elements (like individual text boxes).

MEGA XI does not generally produce output in this format. While I in no way question the integrity of the authors or of the Figure, the format provided diminishes absolute confidence.

General Comments

The authors isolated 3 bacterial strains, from freshwater, from the rhizosphere soil of *Broussonetiae papyrifera*, and from the phycosphere of *Microcystis wesenbergii*. The isolates of the Family Paracoccaceae were characterized genomically and phenotypically. Two of the isolates were designated as the type strains of novel species.

The manuscript is well written and well structured. The methods employed were thorough and appropriate. The focus is interesting and of interest to the general reader.

Have specimens of the newly described species been deposited to two Culture Collections in different countries?

A small number of minor corrections are recommended.

Specific Comments

Line 62: Taxonomic names at the level of Family and below should be italicized, ranks above Family should not be italicized.

Line 70: at the time of review this number is 68.

Line 87: '*Broussonetia*'.

Line 171: 'species'.

Lines 189-90: The phrase 'showing few inconspicuous disparities' is vague and the meaning is unclear.

Line 232: '*Gemmobacter*'.

Line 302: The proposed new species name is '*lacusdiani*': this will need input from a Latin expert but I believe, based on my understanding of the nomenclature rules, that this should be '*lacusdianii*'.

Lines 315 & 339 'pyruvate'.

Line 427 & Figure 1: It is curious and unusual that in the Microsoft Word version of the manuscript. all elements of Figure 1 (species names on each 'leaf', bootstrap values etc.) appear as individual ungrouped elements (like individual text boxes). MEGA XI does not generally produce output in this format. While I in no way question the integrity of the authors or of the Figure, the format provided diminishes absolute confidence.

Re: Comments on Spectrum01088-24 (Extensive genomic study characterizing three *Paracoccaceae* populations and revealing *Pseudogemmobacter lacusdianii* sp. nov. and *Paracoccus broussonetiae* sp. nov.)

Reviewer #1 (Comments for the Author):

The manuscript submitted by Deng Yang, et al. described two new species in family of *Paracoccaceae*, based on phylogenetic, chemotaxonomic and phenotypic analyses. The polyphasic analysis supports the assignment of these strains to be the novel species. Analysis genes associated with key taxonomic characteristics is a good model to merge strain's phenotype and its genome, which will be followed by the other taxonomists.

And also the authors characterizes their abilities of denitrification, MA utilization and PHA biosynthesis. This will supply a deeply understanding on their ecologic roles and potential utilization. I think it is fit to this journal to publishing. While some questions should be discussed or revised before accepted.

Firstly, analysis genes associated with key taxonomic characteristics is very important, authors have clarified the genes related to the ubiquinone, fatty acids and phospholipid PC, PG, while the other lipids also important, especial the diagnostic lipids such as PE, DPG and PME. Whether authors can supply the data of biosynthesis genes related these lipids in part of "Genes associated with key taxonomic characteristics".

Response: We are grateful for this suggestion. The genes related to the biosynthesis of all major lipids (containing PC, PG, PE, DPG, and PME) among three novel strains were determined in their genomes. The results were reorganized to Table S4 and described in part of "Genes associated with key taxonomic characteristics".

Secondary, based genomic and physiologic data, three strains' potential ecological function in original isolation environment should be discussed.

Response: We appreciate your suggestion. The potential ecological function of three novel strains in their original isolation setting has been discussed at the part of Discussion in the revised manuscript, based on the genotypic and phenotypic data.

Some minor revisions should be revised as follow.

1. Line 105, delete "but not in the presence of 2% NaCl" and "but not 40 °C".

Response: We deleted the part that you quoted in this sentence.

2. Data in table S1 is important, so table S1 should be moved to the text, and the same component in Sum in feature lines and the other lines also should be revised or merged.

Response: We accept your suggestion. Accordingly, we moved Table S1 to the main text and renamed as Table 2 in the revised manuscript.

3. Many Abbreviation names were not listed in the Abbreviations part in line 56, please supply them.

Response: Thanks for your reminder, and we listed all of the abbreviation names invovled in the manuscript in the section of Abbreviations.

4. 5-aminolevulinic acid is not a product related wastewater treatment, so suggest move in lines 248-259 to the part of "Secondary metabolite biosynthesis gene clusters analysis".

Response: Indeed, as a precursor for biosynthesis of tetrapyrrole compounds (such as chlorophyll, heme, and vitamin B₁₂), 5-Aminolevulinic acid (5-ALA) has been widely applied in many fields, such as photosensitizers, cosmetics in medicine, plant growth regulators, herbicides and insecticides in agriculture. The research endeavors focused on the microbial synthesis of 5-ALA have received increasing attention due to its dominant benefits over chemical synthesis, such as increased productivity, reduced environmental pollution, and lower economic expenses. With the development of research in this field, many new findings have been reported. Especially, due to the fact that PBS cells could produce 5-ALA during wastewater treatment, there is tremendous promise for combining ALA production with pollutant reductions (Lu et al., 2011 *Bioresour Technol* 1102(20):9503-8; Wu et al., 2010, *Bioresour Technol* 119:55-9; Liu et al., 2014 *Microbiol Biotechnol* 98(17):7349-57). Therefore, we screened the genes related to pathways of wastewater treatment and the biosynthesis of ALA in the genomes of three novel strains and described the results in the section of "Key features involved in wastewater treatment according to the genomic analysis".

5. Deposition certificates of two type strains should be supplied as supplement files.

Response: Yes. We uploaded the certificates from KCTC and CPCC together with the revised manuscript for your check.

Reviewer #2 (Comments for the Author):

General Comments

The authors isolated 3 bacterial strains, from freshwater, from the rhizosphere soil of *Broussonetiae papyrifera*, and from the phycosphere of *Microcystis wesenbergii*. The isolates of the Family *Paracoccaceae* were characterized genomically and phenotypically. Two of the isolates were designated as the type strains of novel species.

The manuscript is well written and well structured. The methods employed were thorough and appropriate. The focus is interesting and of interest to the general reader.

Have specimens of the newly described species been deposited to two Culture Collections in different countries?

Response: We greatly appreciated your time and kind suggestions on help us improve the manuscript.

We uploaded the certificates from KCTC and CPCC together with the revised manuscript for your check.

A small number of minor corrections are recommended.

Specific Comments

Line 62: Taxonomic names at the level of Family and below should be italicized, ranks above Family should not be italicized.

Response: Thank you for your suggestion. According to the International Committee on Systematics of Prokaryotes (ICSP), the taxonomic names from species to phylum should be italicized. Therefore, we italicized all taxonomic names in the revised version.

Line 70: at the time of review this number is 68.

Thank you for reminding us of it. Accordingly, we corrected it in the revised version.

Line 87: 'Broussonetia'.

Response: Corrected.

Line 171: 'species'.

Response: Corrected.

Lines 189-90: The phrase 'showing few inconspicuous disparities' is vague and the meaning is unclear.

Response: We rewrote the related section to make it clear.

Line 232: 'Gemmobacter'.

Response: Corrected.

Line 302: The proposed new species name is 'lacusdiani': this will need input from a Latin expert but I believe, based on my understanding of the nomenclature rules, that this should be 'lacusdianii'.

Response: We totally agree with you. And we corrected to 'lacusdianii' across the full text.

Lines 315 & 339 'pyruvate'.

Response: You are right. We revised the corresponding spelling.

Line 427 & Figure 1: It is curious and unusual that in the Microsoft Word version of the manuscript, all elements of Figure 1 (species names on each 'leaf', bootstrap values etc.) appear as individual ungrouped elements (like individual text boxes). MEGA XI does not generally produce output in this format. While I in no way question the integrity of the authors or of the Figure, the format provided diminishes absolute confidence.

Response: We replaced the corresponding parts in light of your suggestion.

Re: Spectrum01088-24R1 (**Extensive genomic study characterizing three *Paracoccaceae* populations and revealing *Pseudogemmobacter lacusdianii* sp. nov. and *Paracoccus broussonetiae* sp. nov.**)

Dear Dr. Yu-Qin Zhang:

Thank you for the privilege of reviewing your work. Below you will find my comments and instructions from the Spectrum editorial office:

- 1) Why was the KCTC 8066 taxonomically assigned to *Allorhodobacter* sp. in the culture collection? Could this please be corrected?
- 2) Would you have images (preferably scanning or transmission electron microscopy) available of the two new species to add to the manuscript? This is common practice for the description of novel species.

Revision Guidelines

Sincerely,
Eva Sonnenschein
Editor
Microbiology Spectrum

1) Why was the KCTC 8066 taxonomically assigned to *Allorhodobacter* sp. in the culture collection? Could this please be corrected?

When we sent strain CPCC 101601^T to KCTC for deposition, the taxonomic research was still ongoing. However, based on the partial 16S rRNA gene sequence, it showed the closest relationship to members of the family *Paracoccaceae*, with similarities not exceeding 97.4%. Accordingly, we hypothesized that we might have discovered a novel genus within the family *Paracoccaceae* and tentatively proposed the name “*Allorhodobacter* sp.”

We will certainly contact KCTC to update the information on strain CPCC 101601^T after the description of the novel species is accepted for publication. At this stage, no corrections are necessary.

2) Would you have images (preferably scanning or transmission electron microscopy) available of the two new species to add to the manuscript? This is common practice for the description of novel species.

Yes.

We have included transmission electron microscopy images of the cells of these two new species in the newly revised version. Please review them.

Re: Spectrum01088-24R2 (**Extensive genomic study characterizing three *Paracoccaceae* populations and revealing *Pseudogemmobacter lacusdianii* sp. nov. and *Paracoccus broussonetiae* sp. nov.**)

Dear Dr. Yu-Qin Zhang:

congratulations! Your manuscript has been accepted, and I am forwarding it to the ASM production staff for publication. Your paper will first be checked to make sure all elements meet the technical requirements. ASM staff will contact you if anything needs to be revised before copyediting and production can begin. Otherwise, you will be notified when your proofs are ready to be viewed.

Sincerely,
Eva Sonnenschein
Editor
Microbiology Spectrum